# Synthesis and Characterization of Porous, Electro-Conductive Chitosan–Gelatin–Agar-Based PEDOT: PSS Scaffolds for Potential Use in Tissue Engineering

**DOI:** 10.3390/polym13172901

**Published:** 2021-08-28

**Authors:** Dania Adila Ahmad Ruzaidi, Mohd Muzamir Mahat, Zarif Mohamed Sofian, Nikman Adli Nor Hashim, Hazwanee Osman, Mohd Azizi Nawawi, Rosmamuhamadani Ramli, Khairil Anuar Jantan, Muhammad Faiz Aizamddin, Hazeeq Hazwan Azman, Yee Hui Robin Chang, Hairul Hisham Hamzah

**Affiliations:** 1Faculty of Applied Sciences, Universiti Teknologi MARA, Shah Alam 40450, Malaysia; dniahmad1998@gmail.com (D.A.A.R.); azizi_nawawi@uitm.edu.my (M.A.N.); rosma614@uitm.edu.my (R.R.); khairil0323@uitm.edu.my (K.A.J.); faizaizamddin@gmail.com (M.F.A.); 2Department of Pharmaceutical Technology, Faculty of Pharmacy, Universiti Malaya, Kuala Lumpur 50603, Malaysia; 3Institute of Biological Sciences, Faculty of Science, Universiti Malaya, Kuala Lumpur 50603, Malaysia; nikmanadli@um.edu.my; 4Centre for Drug Research in Systems Biology, Structural Bioinformatics and Human Digital Imaging (CRYSTAL), Universiti Malaya, Kuala Lumpur 50603, Malaysia; 5Centre of Foundation Studies UiTM, Universiti Teknologi MARA (UiTM), Cawangan Selangor, Kampus Dengkil, Dengkil 43800, Malaysia; hazwanee@uitm.edu.my; 6Centre for Foundation and General Studies, Universiti Selangor, Bestari Jaya 45600, Malaysia; hazeeq87@unisel.edu.my; 7Faculty of Applied Sciences, Universiti Teknologi MARA, Cawangan Sarawak, Samarahan 94300, Malaysia; robincyh@uitm.edu.my; 8School of Chemical Sciences, Universiti Sains Malaysia, Gelugor 11800, Malaysia

**Keywords:** chitosan, PEDOT: PSS, scaffold, electrical conductivity, tissue engineering

## Abstract

Herein we report the synthesis and characterization of electro-conductive chitosan–gelatin–agar (Cs-Gel-Agar) based PEDOT: PSS hydrogels for tissue engineering. Cs-Gel-Agar porous hydrogels with 0–2.0% (*v*/*v*) PEDOT: PSS were fabricated using a thermal reverse casting method where low melting agarose served as the pore template. Sample characterizations were performed by means of scanning electron microscopy (SEM), attenuated total reflectance–Fourier transform infrared spectroscopy (ATR–FTIR), X-ray diffraction analysis (XRD) and electrochemical impedance spectroscopy (EIS). Our results showed enhanced electrical conductivity of the cs-gel-agar hydrogels when mixed with DMSO-doped PEDOT: PSS wherein the optimum mixing ratio was observed at 1% (*v*/*v*) with a conductivity value of 3.35 × 10^−4^ S cm^−1^. However, increasing the PEDOT: PSS content up to 1.5 % (*v*/*v*) resulted in reduced conductivity to 3.28 × 10^−4^ S cm^−1^. We conducted in vitro stability tests on the porous hydrogels using phosphate-buffered saline (PBS) solution and investigated the hydrogels’ performances through physical observations and ATR–FTIR characterization. The present study provides promising preliminary data on the potential use of Cs-Gel-Agar-based PEDOT: PSS hydrogel for tissue engineering, and these, hence, warrant further investigation to assess their capability as biocompatible scaffolds.

## 1. Introduction

The therapeutic potential of scaffolds for tissue engineering mainly hinges on finding the right biomaterials that are capable of restoring, retaining and revitalizing damaged tissues at the site of injury [1,2]. In view of this, the use of conductive scaffolds has been considered to be one of the promising options to promote new tissue morphogenesis [3]. Their intrinsic conductance permits signal transmission, which facilitates the alignment of cells into specific directions and further leads to differentiation of cells into various tissue types [4]. For this reason, many conductive scaffolds have been developed by incorporating a variety of conductive components such as metals, ions, carbon nanomaterials and conducting polymers (CPs) [5,6,7,8,9,10,11].

Among them, CPs such as polypyrrole (PPy), polyaninline (PANi) and poly (3,4-ethylenedioxythiophene) (PEDOT) offer tremendous advantages, as they are easy to functionally modify at the molecular level [12,13,14,15]. Even though conductive scaffolds may offer a more significant clinical impact than non-conductive scaffolds, major setbacks including poor biodegradability, unknown long-term toxicity and non-homogenous distribution of the conductive particles in the composite assembly remain the limiting factors to their successful use [16,17,18,19,20,21,22]. 

Hydrogel scaffolds have emerged as one of the most extensively studied biomaterials in this area due to their unique, compositional, highly tunable functionalities and structural similarities that mimic the extracellular matrix [23,24,25]. These 3-D polymeric structures are normally constructed by crosslinking hydrophilic polymeric chains [26,27,28,29]. Depending on the polymeric composition, hydrogels can be categorized into natural, synthetic and hybrid hydrogels [30,31,32,33,34]. While the latter two types are normally associated with low biocompatibility in vitro and in vivo [35,36], batch variations, limited functionality and potential immunogenic effects limit the applications of the natural types of hydrogels [30,37,38].

The selection of (i) proper techniques and (ii) the right combination of biomaterials hence dictates the success of producing scaffold matrices that are not only electroactive, but also possess excellent biocompatibility and functionality. Apart from the above-mentioned properties, the porosity of the scaffolds has been also reported to play a significant role in the success of conductive scaffolds in these roles [39,40,41].

Based on this information, the present study describes the fabrication of newly engineered, porous, electroactive hydrogel scaffolds using chitosan and gelation as a base. Chitosan is one of the natural polymer derivatives commonly used in scaffold fabrication due to its favorable properties for wound healing. It is widely used in various biomedical fields, especially in tissue engineering, antimicrobial activity, cancer treatment and drug delivery [17]. Other biomaterials such as gelatin have potential benefits for pharmaceutical applications due to their high biocompatibility and biodegradability, and they are also less antigenic [6,40]. The selection of the base materials was because these materials could provide a supportive environment for tissue regeneration due to their supreme physicochemical features [42,43]. 

Agarose with a low melting temperature was used in this study to serve as a sacrificial template for the pore-making purpose of hydrogels [44,45]. To render these hydrogels electrically conductive, we incorporated PEDOT: poly (4-styrenesulfonic acid) (PSS) into them. Due to proper bond length and chemical properties, PEDOT: PSS offers greater chemical and thermal stability than other commonly used CPs including Ppy and PANi [46,47,48,49,50,51]. PSS was added to improve the aqueous solubility and to act as a primary dopant to PEDOT [52]. It is worth noting that the scope of this study was only limited to the synthesis and characterization of the hydrogels wherein different key analyses were performed to characterize their physicochemical properties. Based on the results that we obtained, the potential use of these fabricated hydrogels in tissue engineering is discussed in the later part of this manuscript. 

## 2. Materials and Methods

### 2.1. Materials 

3,4-Ethylenedioxythiophene (EDOT) (Mw = 142.18), poly (4-styrenesulfonic acid) (PSS) solution (Mw ~75,000, 18 wt.% in H_2_O), ammonium persulfate ((NH_4_)_2_S_2_O_8_) (reagent grade, 98%), dimethyl sulfoxide (DMSO) (ACS reagent, >99.9%), low molecular weight chitosan (50,000–190,000 Da) (based on viscosity), glutaraldehyde (grade II, 25% in H_2_O) and acetic acid (reagent plus, >99%) were purchased from Sigma Aldrich (St. Louis, MO, USA). Gelatin powder (pure, Ph. Eur.) and agarose (medium EEO) were supplied by Fluka (Munich, Germany). 

### 2.2. Synthesis of Conductive PEDOT: PSS

Conductive PEDOT: PSS solution was prepared using a chemical oxidative polymerization method [53]. Briefly, 0.05 mL of hydrophobic EDOT was added to 4.50 mL deionized water together with 0.08 mL hydrophilic PSS. The solution was left, stirring, for 60 min at room temperature to obtain a solution of EDOT: PSS. Next, 0.55 g of ammonium persulfate, which acted as an oxidizing agent, was added into the solution to initiate the polymerization of EDOT. A dark blue solution was obtained after 120 min of stirring, indicating that the polymerization of PEDOT had occurred. Another 4.50 mL of deionized water was added into the PEDOT: PSS solution and was left, stirring, for another 30 min to complete the polymerization of EDOT radical cations. Exactly 3.0 vol.% of DMSO was subsequently added into the solution to make the PEDOT: PSS more electrically conductive prior to stirring for another 120 min. 

### 2.3. Fabrication of Porous Hydrogels 

The chitosan–gelatin (Cs–Gel)-based PEDOT: PSS porous hydrogel scaffolds were fabricated using a reversed casting method. Briefly, this method was composed of three main steps: (i) solution mixing, (ii) freezing process and (iii) removal of agarose (Table 1).

(i)Solution mixing: Exactly 7.2 wt.% of gelatin powder was added into 7% acetic solution followed by the addition of 1.8 wt.% of chitosan. The mixture was stirred for 30 min at 80 °C. After the mixture was completely dissolved and formed a yellowish clear solution, exactly 0.05 wt.% agarose was added to regenerate pores and was left, stirring, for another 30 min at 90 °C. Consequently, a specific volume of DMSO–PEDOT: PSS was added into the solution, as shown in Table 1.(ii)Freezing process: Exactly 0.5% glutaraldehyde was added into the Cs–Gel–Agar–PEDOT: PSS solution, and the resulting mixture was left for 30 min at room temperature. This addition was done to establish the chemical crosslinking of the molecules present in the Cs–Gel–Agar-PEDOT: PSS solution. Next, the mixture underwent a freezing process at −20 °C for 180 min to form hydrogels.(iii)Removal of agarose: The agarose was used as a sacrificial template for fabricating the porous-structured hydrogels. The removal of agarose was done by immersing the hydrogels in an extraction bath containing 10 *v*/*v* % glutaraldehyde at 90 °C. Figure 1 shows the fabricated porous hydrogels.

### 2.4. Characterization of Porous Hydrogels

#### 2.4.1. Morphology and Pore Size Observations 

The morphology and the structure of the porous hydrogels were investigated using scanning electron microscopy (SEM). All samples (*n* = 5) underwent a drying process (75 °C, 30 min) to remove moisture and undergo a gold-sputtering step. Next, the morphology images were analyzed using ImageJ software. 

#### 2.4.2. Attenuated Total Reflectance–Fourier Transform Infrared (FTIR) Analysis

The FTIR spectra of all samples were recorded within the range of 4000 to 550 cm^−1^ using a Perkin-Elmer spectrometer. The resolution was set at 8 cm^−1^, and 4 scans were performed for each measurement. Prior to sample measurement, a background spectrum was recorded and automatically subtracted from the spectra of the sample. All spectra were baseline corrected. Spectral analysis was performed using Spectrum software (version 10, Perkin-Elmer, Ltd., London, United Kingdom) to elucidate the possible chemical associations between the components within the hydrogel assembly.

#### 2.4.3. X-ray Diffraction (XRD) Analysis

The X-ray diffraction (XRD) pattern of the porous hydrogels was obtained using a PANalytical X-Pert Pro XRD machine. The parameters were set with CuKα radiation of wavelength 1.541 Å, scanning angle 2*θ* over a range of 5–90°, step size 0.02/s, 45 kV voltages and 40 mA current. The XRD pattern was analyzed using “X’pert Highscore” and “X’pert Plus” software (brand) to further identify the properties of the porous hydrogels. 

#### 2.4.4. Electrochemical Impedance Spectroscopy (EIS) Analysis

The conductivity of the hydrogel solutions with the addition of different wt. % PEDOT: PSS was determined using HIOKI 3520 LCR Hi-Tester electrochemical impedance spectroscopy (EIS) in a frequency range of 100 Hz to 10,000 Hz at room temperature. A cylindrical probe was used to detect the charge flow in the solutions. For every operating frequency, the impedance was obtained accordingly. Consequently, the values of impedance were used in the calculations to determine the conductivity values of scaffold samples.

#### 2.4.5. In Vitro Stability Studies

To study the chemical stability of the porous hydrogels, the hydrogels with 1 g weight were submerged in test tubes containing 10 mL of phosphate-buffered saline (PBS) solution at pH 7.4 and maintained at 37 °C to mimic physiological conditions. One set (*n* = 5) of samples was removed after 30 min incubation and underwent a rinsing process to remove the excess PBS salts left in the hydrogels. An ATR–FTIR analysis was performed to determine the hydrogels’ chemical stability performances.

## 3. Results and Discussion

### 3.1. Porous Hydrogel Formation

(a)Morphology of Porous Hydrogels formed from the Reversed Casting Method

We analyzed the output generated from the SEM to determine the morphology of the porous hydrogels and to estimate the pore sizes formed from the reversed casting method. The SEM micrograph of Figure 2a shows a cross-section of the hydrogels’ outer layer, while Figure 2b shows a cross-section of the hydrogels’ inner structure. The size of the pores in Figure 2a was found to be smaller than in Figure 2b. This may have been due to the drying process of the hydrogels, in which the outer layer of the hydrogels comparatively received more heat, causing them to shrink. 

Figure 2c illustrates the process for the removal of agarose for pore-making purposes. In this process, we mixed low-melting-temperature agarose into a hydrogel mixture. The hydrogels were then immersed in an extraction bath to allow and facilitate the removal of agarose. The denatured agarose slowly moved out of the hydrogels and left the pores due to the weak hydrogen bonds formed between the agarose and the other molecules present in the mixture. 

(b)Pore Size Observation of Porous Hydrogels

According to the SEM micrographs of all samples in Figure 3, we can see that the hydrogels’ pore diameters without the addition of PEDOT: PSS are larger compared to the other porous hydrogels. This is due to the viscosity of PEDOT: PSS dispersion, which blocks the pores, causing a reduction in pore sizes [53]. Physically, this reversed casting method of fabrication created a spongy resistance level in the porous hydrogels. The obtained images were analyzed with ImageJ software (NIH) to assess the pore diameter of the porous hydrogels (Figure 4). Generally, the average pore diameter was within the range of 703 µm to 1659 µm. As shown in Table 2, the different volume ratios of PEDOT: PSS in this study did not play an important role in obtaining hierarchically porous hydrogels as had been suggested in a previous study [54]. Due to its long-chain structure and specific orientation after crystallization, adding PEDOT: PSS can reduce the porosity of hydrogels, thus decreasing the density of conductive paths [54].

### 3.2. Proposed Mechanism for the Chemical Reaction of Porous Hydrogels

#### 3.2.1. Synthesis of Conductive PEDOT: PSS-Doped DMSO Dispersion

In this section, we propose a mechanism that could explain the synthesis and doping processes of PEDOT: PSS, according to previous studies [52,55]. In these studies, ammonium persulfate (APS) was used as the oxidant for the previously mentioned purpose. The first step involved the dissociation of APS, which led to the production of two ammonium ions (NH_4_^+^) and one persulfate ion (S_2_O_8_^2−^) in the presence of water (Equation (1)). The persulfate ion underwent homolytic fission to produce sulfate radicals [SO_4_^•−^]: (NH_4_)_2_ S_2_O_8_ (aq) → 2NH_4_^+^ (aq) + S_2_O_8_^2−^ (aq)S_2_O_8_^2−^ (aq) ⇌ 2SO_4_^•−^ (aq) (1)

The initiation step involved the oxidation of EDOT by sulfate radicals to produce EDOT radical anions and sulfate cations. In the propagation step, two EDOT radical anions interacted and coupled to each other, followed by the elimination of a pair of hydrogen molecules to produce the PEDOT dimer. The PEDOT dimer was oxidized again in the same manner as the EDOT to produce a dimer radical anion. Next, the dimer radical anion propagated into a longer chain by reacting with another EDOT radical anion to produce PEDOT. For primary doping in the presence of ammonium ions and sulfate ions, the positive thiophene sites of hydrophobic PEDOT tend to react with the negative sulfonic sites of hydrophilic PSS by creating new covalent bonds (Figure 5). 

DMSO is a polar so-solvent and has been selected as a secondary dopant to PEDOT: PSS due to its high dipole movement of molecules, which creates dipole–charge interactions between PEDOT: PSS and DMSO [56,57]. This, then, leads to high charge carrier mobility. A good secondary dopant such as sulfuric acid, sulfonic acid or sulfoxide derivatives must have good diffusibility into the PSS shell [58]. Generally, a solvent that is a good candidate for increasing PEDOT: PSS conductivity is from a polar group of solvents that strongly interact with the sulfonic group of PSSH. DMSO is known to be a strong polar co-solvent that is able to permanently increase the conductivity of PEDOT: PSS by facilitating the growth and interconnection of PEDOT-rich domains. This indirectly causes electron transfer to become more effective [59]. As shown in Figure 6, PEDOT grains (black) are surrounded by an insulating PSS shell (purple). After the addition of DMSO, the PSS shell layer becomes thinner and thus facilitates the transportation of charges along the PEDOT chain. This phenomenon also results in a higher charge transportation rate than the undoped PEDOT: PSS and therefore increases the conductivity value of PEDOT: PSS [55].

The spherical nanostructure of PEDOT: PSS elongates into an ellipsoidal form after the addition of DMSO [60]. The DMSO addition causes further elongation of the compact coil of PEDOT: PSS (Figure 6) as well as facilitating the transportation of charges due to such elongation. These activities allow the localized electronic states of PEDOT: PSS to become partially localized and enhance the mobility of charge carriers [59]. In some cases, DMSO was removed through an annealing process to obtain pure PEDOT: PSS hydrogels. Surprisingly, the interconnecting effect from the DMSO dopant remained the same. This explains that DMSO causes permanent conformational changes toward the PEDOT: PSS system. Previous studies have reported that DMSO is able to increase the conductivity properties of PEDOT: PSS hydrogels [61,62]. Pasha et al. proved that 20 wt.% of DMSO yields high conductivity (110 Scm^−1^ > σ< 140 Scm^−1^) [61], while Fadhil and Setia (2018) discovered that conductive PEDOT: PSS film doped with 75% DMSO on polyethylene terephthalate (PET) yielded high conductivity (770 Scm^−1^) [63]. Through these findings, it can be said that the use of different DMSO wt.% gives relatively different conductivity effects. 

#### 3.2.2. Possible Reactions of Chitosan–Gelatin–Agar-Based PEDOT: PSS Porous Hydrogel Scaffolds

The porous hydrogels were made from three main materials, which were chitosan, gelatin and PEDOT: PSS. Although the agarose had been removed, the possibility that excess agarose remained in the porous hydrogels still existed. In this section, we illustrate how these three materials interact among themselves. Figure 7, Figure 8 and Figure 9 show three possible reactions that may occur. Figure 7 shows the interaction between the hydroxyl group of agaroses and the ether group of chitosan. A new carbon–carbon bonding formed between them by removing water (H_2_O) molecules. Figure 8 shows the formation of a new covalent bond between nitrogen from the amino group of chitosan and oxygen from the carboxyl group of gelatin. Meanwhile, Figure 9 shows the formation of the S–N bond of sulfenamide, made between the amino group of chitosan and the thiophene ring of PEDOT. These explain that the active sites of chitosan tend to form a bond with all materials present in a hydrogel and thus, enhance molecular bonding. 

Meanwhile, Figure 10 shows possible reactions that may occur in the hydrogels after the crosslinking process using glutaraldehyde. To the best of our knowledge, the crosslinking process only affects the active sites of gelatin and chitosan. As depicted in Figure 10a, the presence of lone pair electrons from nitrogen atoms causes the nucleophilic attack on the carbonyl carbons of glutaraldehyde [63]. Conjugated chitosan–glutaraldehyde is connected through the covalent bonds formed [21]. This reaction concept can also be applied to the gelatin–glutaraldehyde interaction (Figure 10b). The consideration that glutaraldehyde may be bonded to both gelatin and chitosan at one time is also illustrated (Figure 10c). Physically, the crosslinked hydrogels are tougher due to compacted molecular structures inside them [33,64,65].

### 3.3. Attenuated Total Reflectance–Fourier Transform Infrared Spectroscopy (ATR–FTIR) of the Porous Hydrogels

In order to study the DMSO doping process and the interaction of PEDOT: PSS with hydrogels, ATR–FTIR was performed. Figure 11 shows the ATR–FTIR plot of pure PEDOT: PSS and doped PEDOT: PSS. C-H and O-H stretching of PSS were detected at ~3733 to 3422 cm^−1^. O-H stretching could also be detected at ~2254 cm^−1^ [66]. The C-S bond in the thiophene rings of PEDOT: PSS was detected, with a peak at ~720 cm^−1^. Peaks at both ~1085 cm^−1^ and ~1197 cm^−1^ indicated the presence of S-O symmetric and S-Phenyl bonds [51,61]. With the addition of DMSO as a secondary dopant, the S-O symmetric bond became slightly intense and could be detected at ~1036 cm^−1^. The CH_3_ bond of the DMSO was detected at ~1400 cm^−1^ [67,68].

The ATR–FTIR analysis was conducted on hydrogel samples to compare the functional groups and bonds present with our proposed possible reactions (refer to Section 3.2). Figure 12 shows the ATR–FTIR plot of specific peaks that could be used to differentiate the samples, while Figure 13 shows the ATR–FTIR plotting of all fabricated porous hydrogels. In Figure 12: (i) Overall, a wider peak from ~3100 cm^−1^ to ~3400 cm^−1^ represents the presence of C-H and O-H bonds. There is a partial overlap of amine and hydroxyl group stretching vibrations between wavelengths of ~3200 cm^−1^ and ~3400 cm^−1^. (ii) The detection of functional groups after the addition of agar and PEDOT: PSS was problematic due to the overlapping of various functional groups. However, we did some comparisons between doped PEDOT: PSS and Cs–Gel–Agar-PEDOT: PSS to detect any possible changes. As illustrated by the graph in Figure 11, the intensities of peaks from wavenumbers ~1100 cm^−1^ to ~1400 cm^−1^ became flatter as the doped PEDOT: PSS was mixed into the hydrogel solution. This scenario may be attributed to the breaking of bonds of more thiophene groups to form new bonds with the amide groups present in chitosan and gelatin. (iii) Roughly, we can see that the plotting of hydrogels with glutaraldehyde (Glu) recorded an intensified graph compared to other graphs. This suggests that glutaraldehyde as a crosslinker could react with an amide group of chitosan and gelatin and transform into a new conjugated structure through a Schiff base reaction [33,69]. (iv) Lastly, we compared the gelatin curve before and after the addition of chitosan. The N-H bond (~1650 cm^−1^) and acetyl group (~1260 cm^−1^) were related to the remaining chitin inside the chitosan polymer. C-O-C stretching, the vibrational mode of amide (~1592 cm^−1^ to ~1602 cm^−1^) and the C-O stretch of the primary alcoholic group (~1370 cm^−1^) indicate peaks that belong to chitosan [70,71]. 

### 3.4. Phase and Crystallinity Studies of Porous Hydrogels

The XRD pattern of all porous hydrogels was compared with the Cs–Gel (control) sample, as shown in Figure 14. According to Sakunpongpitiporn (2019), two main peaks at 17.7° and 25.8° are referred to as the amorphous halo diffraction of PSS and the interchain packing of PEDOT, respectively [52]. In our case, the presence of PEDOT: PSS in the Cs–Gel hydrogel could be clearly detected in the Cs–Gel–1.0% PEDOT: PSS sample (16.65° and 22.58°). The shifting of peaks might have been due to the conjoining of the chitosan–gelatin mixture with the PEDOT: PSS dispersion. Other samples did not show any significant crystalline peaks because of the amorphous structures of the hydrogels [72]. The decreased intensity of PEDOT: PSS at 1.5% and 2.0% indicated a reduction in the degree of crystallinity of the composite scaffolds due to the incorporation of and interactions among gelatin, chitosan and PEDOT: PSS [73]. 

### 3.5. Electrical Conductivity of Porous Hydrogels

PEDOT: PSS is a well-known conducting polymer in addition to polyaniline, polypyrrole and polythiophene [46,47]. The presence of free-moving electrons along the PEDOT chain allows charge transfer. This scenario causes PEDOT: PSS to be conductive. In our study, the conductivity value of pure PEDOT: PSS increased to 3.75 × 10^1^ S m^−1^ with the addition of an organic solvent, DMSO (Table 3). Because semiconductor materials have reasonable conductivity values, ranging from 10^−5^ to 10^1^ S m^−1^, we managed to synthesize a conductive PEDOT: PSS with a conductivity value within the range of typical semiconductor materials. 

As shown in Table 3, the conductivity values of the hydrogel solutions increased as the volume (%) of PEDOT: PSS increased. The highest conductivity value was given by a 1.0% PEDOT: PSS addition. The conductivity value recorded a slight drop pattern with the addition of 1.5% and 2.0% PEDOT: PSS. This indicated that the addition of 1.0% PEDOT: PSS was the optimal vol.% required to achieve the highest conductivity value. The decline in conductivity values starting with 1.5% PEDOT: PSS content might have been due to the strong interaction between insulating chitosan and conducting PEDOT. The amine group from chitosan tends to make bonds with the thiophene group of PEDOT. This causes a disturbance along the PEDOT backbone and hence, lowers the effectiveness of charge mobility. 

We then compared the conductivity of hydrogels with and without the presence of agarose. As shown in Figure 15, the presence of agarose contributed to a higher electrical conductivity value compared to the values without agarose present. Previous findings have reinforced the understanding that the addition of agarose can increase the electrical conductivity value of composite scaffolds, but it may not be able to provide any immune response [48]. This drawback could be one of the prime reasons that discourages the use of agarose in tissue engineering. Even so, agarose can still be used in producing composite scaffolds, as its lack of an immunogenicity response can be overcome by the presence of other materials. The utilization of agarose to increase a composite’s conductivity could be a better solution compared to other materials, whose biocompatibility is still in doubt. Nevertheless, studies on the mechanisms of electrical conductivity in agarose are still in their infancy, with more work required to understand the underlying mechanisms in more detail and under different circumstances. In addition, this study demonstrated that hydrogels with 1.0% PEDOT: PSS exhibited the highest conductivity (3.35 × 10^−2^ S m^−2^) (Table 4). When considering the reducing effects of inserting conductive hydrogels under physiological conditions on PEDOT: PSS’ initial conductivity value, it is worth noting that this is still within the range of cortical bone conductivity (0.02 S m^−1^) [49]. Thus, we believe that these findings have provided further evidence to support the use of PEDOT: PSS for improving bone grafting technology [50].

### 3.6. Stability of Porous Hydrogels under PBS Conditions

In order to investigate the stability of the porous hydrogels, an in vitro PBS test was conducted. All samples (*n* = 5) were submerged in 10 mL of 7.4 pH PBS solution for 30 min. Figure 16 shows the physical appearance of each sample after the immersion process. From our observations, there were no significant differences in the hydrogels after 30 min immersion in terms of their color intensity. We further explored the hydrogels’ stability through ATR–FTIR characterization to analyze the chemical degradation of porous hydrogels. As shown in Figure 17, samples (d) and (e), with 1.5% and 2.0% PEDOT: PSS content, respectively, show characteristic peaks of PEDOT: PSS. Peaks from ~1300 cm^−1^ to ~1650 cm^−1^ represent the C-O-C, C-C and C=C bonds of PEDOT and PSS [63]. However, the S-O symmetric bond of PEDOT could not be detected. PEDOT might have been degraded in the PBS due to PEDOT leaching out from the hydrogels. According to the previous literature, materials that degrade too fast are not useful, while a moderate rate of degradation is suitable for tissue engineering applications [74]. This problem should be highlighted in future studies.

## 4. Conclusions

In this study, we showed evidence to support the use of PEDOT: PSS in fabricating conductive scaffolds to support the growth of electrically responsive cells and tissues. We also successfully synthesized conductive PEDOT: PSS dispersion with the use of DMSO as a secondary dopant via the chemical oxidative polymerization method. This was because the DMSO could facilitate the growth and interconnection of PEDOT-rich domains, resulting in an increase in the conductivity of the synthesized, porous, structured hydrogels through an effective electron transfer within the scaffold structure. In addition, we demonstrated a facile approach to using low-melting-temperature agarose when fabricating porous, structured hydrogels as well as the use of glutaraldehyde crosslinker spectroscopically for enhancing molecular bonding. Based on our findings, we highly recommend the use of PEDOT: PSS for improving the electrical conductivity performance of porous hydrogels. We managed to produce a porous hydrogel scaffold with an electrical conductivity value on par with cortical bone tissues. The in vitro PBS study emphasized that the ability of PEDOT: PSS to be stable in porous hydrogels was problematic and should be further explored. These findings warrant investigations in terms of the scaffolds’ stability, biodegradability, and toxicity to provide a well-rounded approach to improving scaffold-grafting technology. 

## Figures and Tables

**Figure 1 polymers-13-02901-f001:**
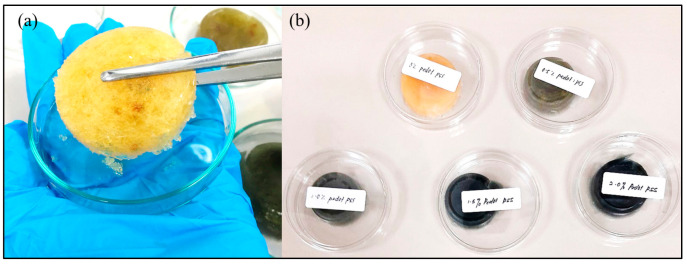
(**a**) Close-up image of porous hydrogel; (**b**) scaffold samples after removal of agarose.

**Figure 2 polymers-13-02901-f002:**
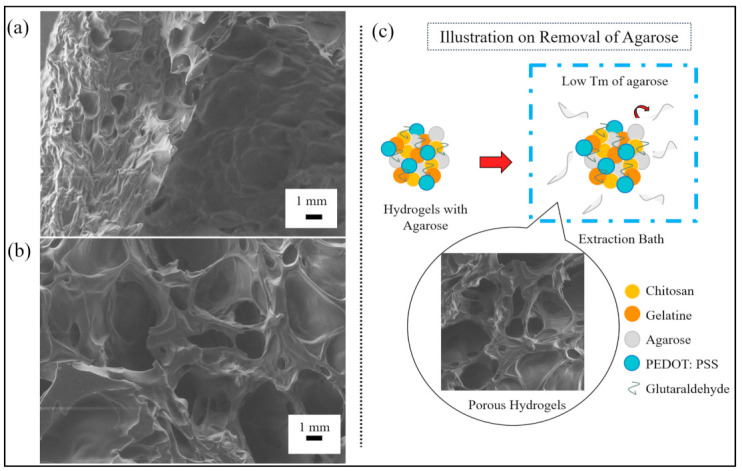
SEM micrographs of a porous hydrogel’s (**a**) outer layer structure and (**b**) inner layer structure. (**c**) An illustration of hydrogels undergoing the removal of agarose.

**Figure 3 polymers-13-02901-f003:**
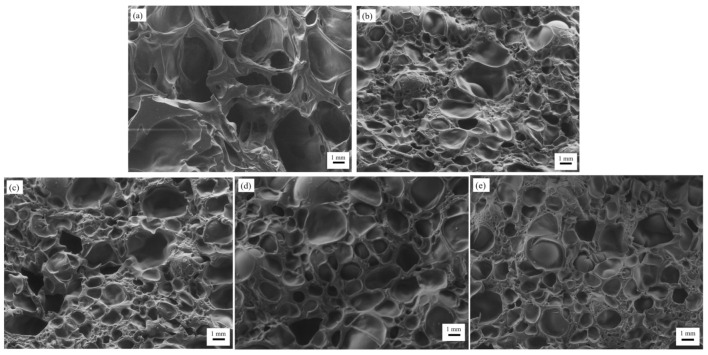
SEM micrographs of all samples: (**a**) Cs–Gel (control sample), (**b**) Cs–Gel–0.5% PEDOT: PSS, (**c**) Cs–Gel–1.0% PEDOT: PSS, (**d**) Cs–Gel–1.5% PEDOT: PSS and (**e**) Cs–Gel–2.0% PEDOT: PSS.

**Figure 4 polymers-13-02901-f004:**
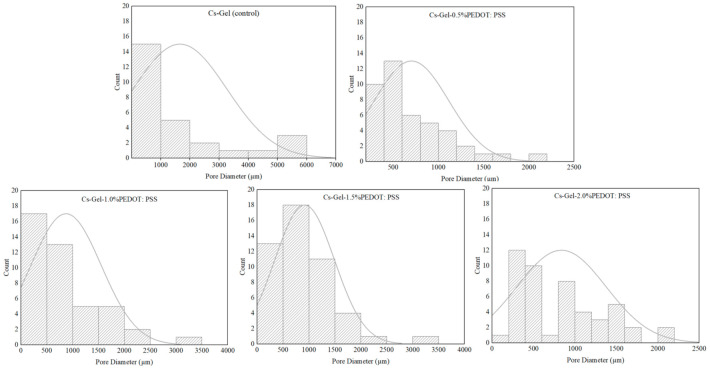
Pore diameter distribution graphs of all samples, analyzed via ImageJ software.

**Figure 5 polymers-13-02901-f005:**
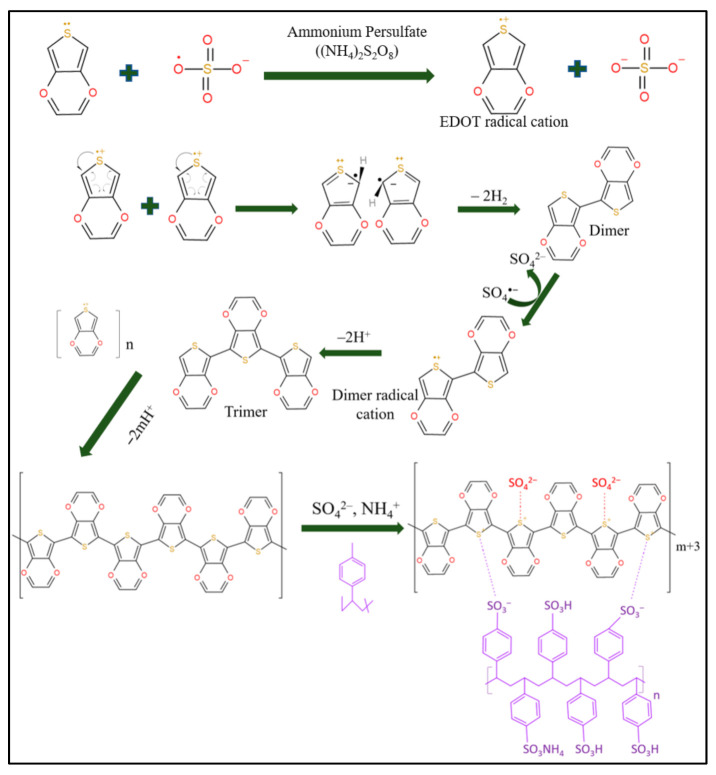
The oxidation step of the EDOT monomer; the propagation step, when the EDOT dimer becomes an EDOT trimer through the formation of a dimer radical cation; and a primary doping illustration of PEDOT: PSS.

**Figure 6 polymers-13-02901-f006:**
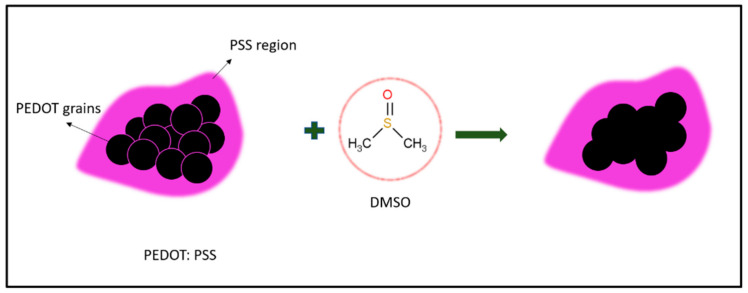
Illustration of secondary doping of PEDOT: PSS using DMSO organic solvent.

**Figure 7 polymers-13-02901-f007:**
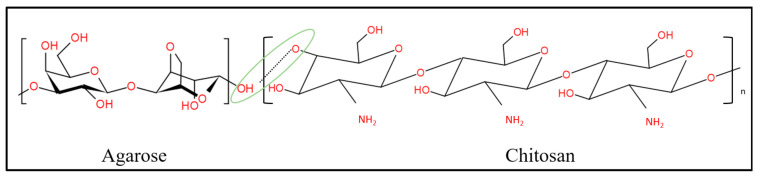
Chemical interaction between the hydroxyl group of agaroses and the ether group of chitosan.

**Figure 8 polymers-13-02901-f008:**
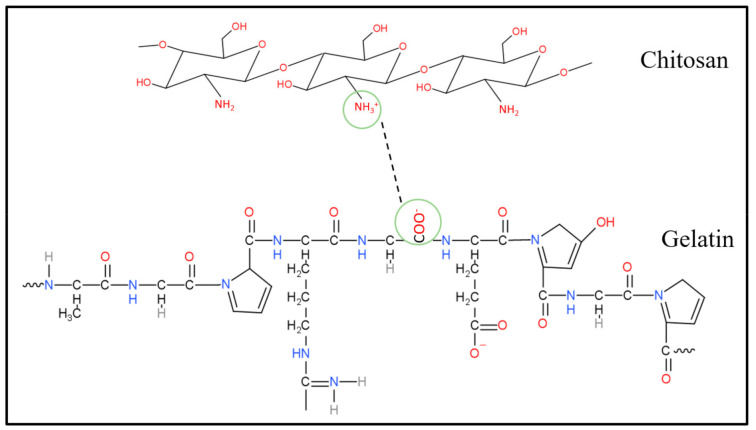
Chemical interaction between the amino group of chitosan and a carboxyl group of gelatin.

**Figure 9 polymers-13-02901-f009:**
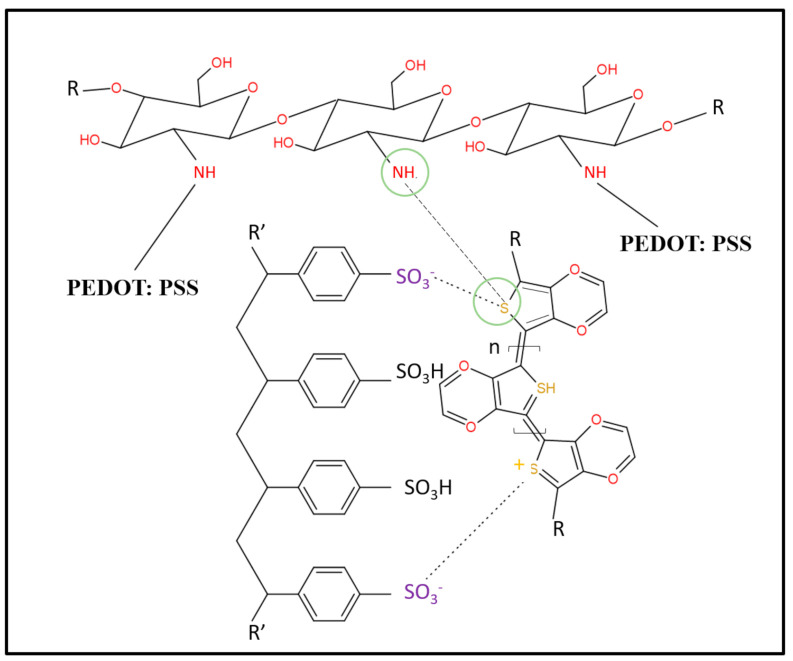
Chemical interaction between the amino group of chitosan and the thiophene ring of PEDOT.

**Figure 10 polymers-13-02901-f010:**
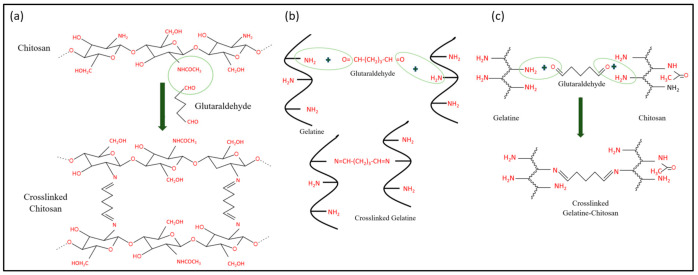
(**a**) Crosslinking interaction between chitosan and glutaraldehyde; (**b**) crosslinking interaction between gelatin and glutaraldehyde; and (**c**) crosslinking interaction between chitosan, glutaraldehyde and gelatin.

**Figure 11 polymers-13-02901-f011:**
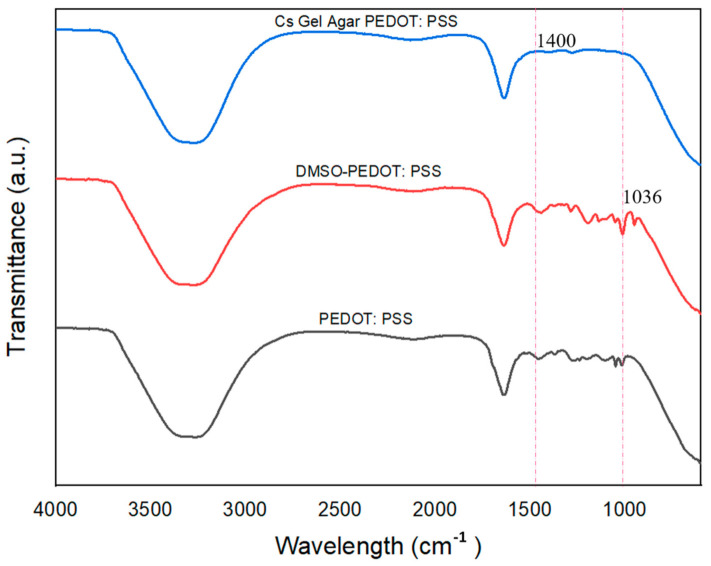
The ATR-FTIR plots of pure PEDOT: PSS, PEDOT: PSS-doped DMSO and Cs–Gel–Agar–PEDOT PSS.

**Figure 12 polymers-13-02901-f012:**
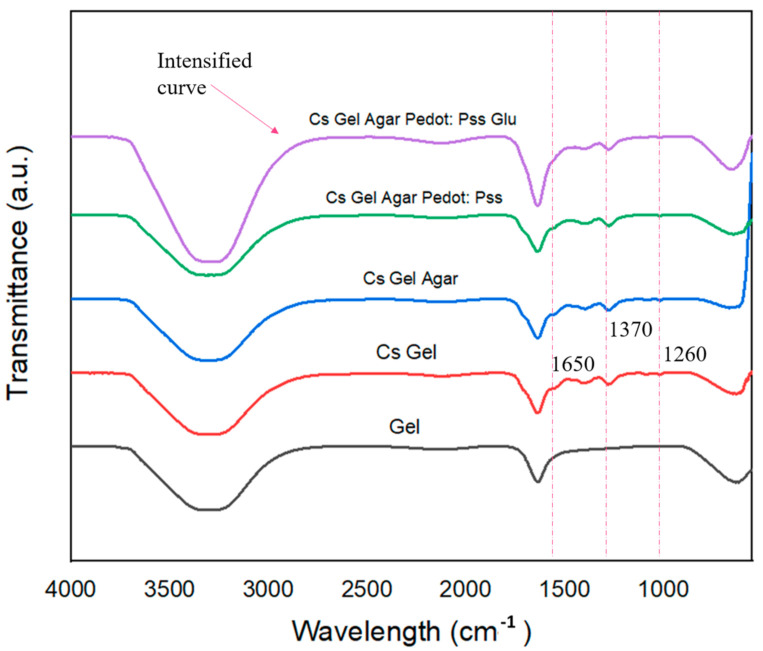
ATR–FTIR plots of hydrogel solutions.

**Figure 13 polymers-13-02901-f013:**
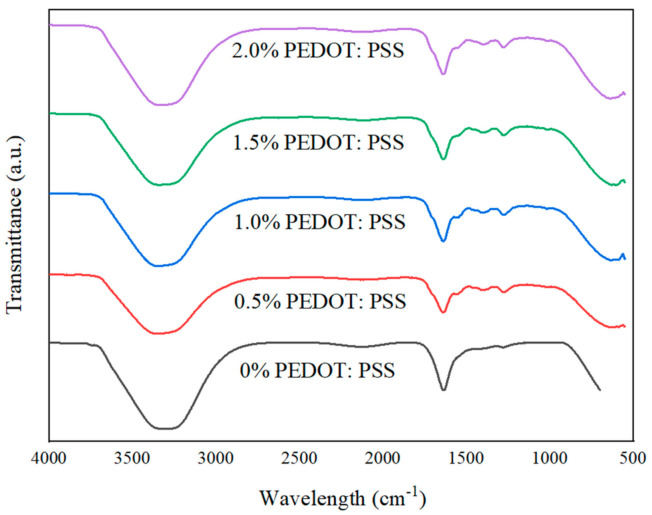
ATR–FTIR plots of fabricated porous hydrogels.

**Figure 14 polymers-13-02901-f014:**
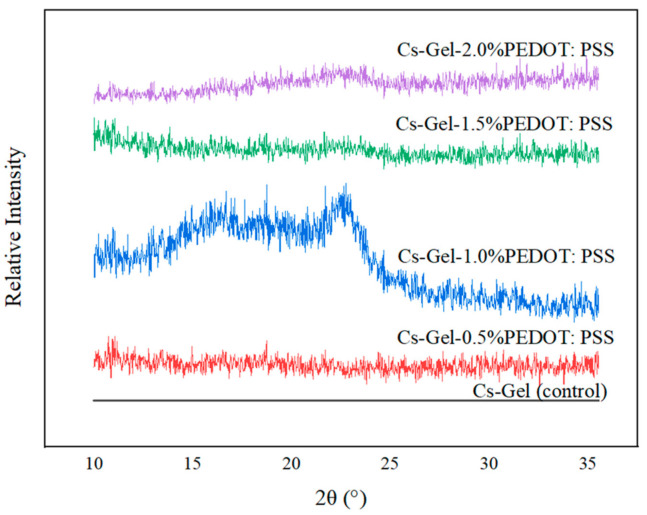
Relative intensity XRD plot of porous hydrogels.

**Figure 15 polymers-13-02901-f015:**
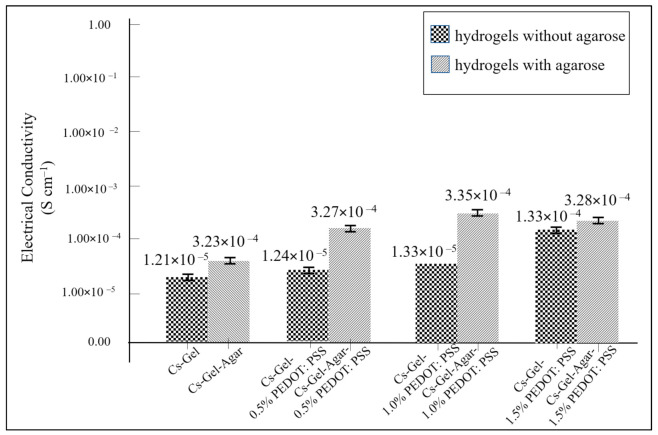
The electrical conductivity plots of all samples.

**Figure 16 polymers-13-02901-f016:**
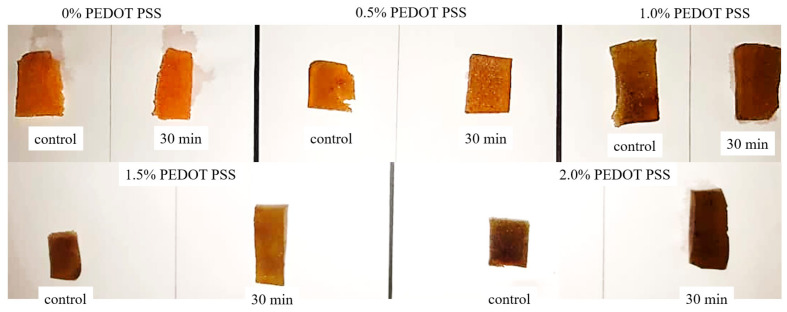
The physical appearance of all fabricated hydrogels before and after the immersion process in PBS solution.

**Figure 17 polymers-13-02901-f017:**
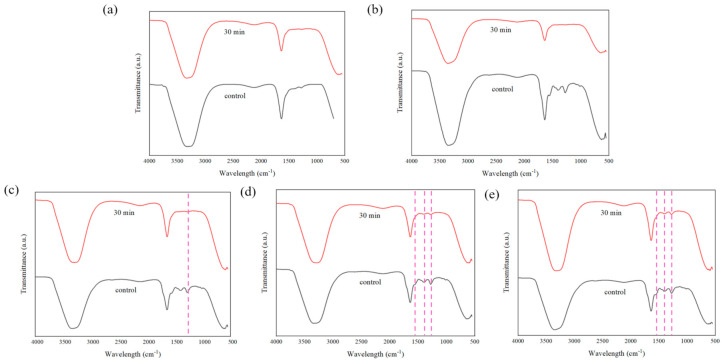
ATR–FTIR plots of fabricated porous hydrogels after 30 min immersion in PBS solution where (**a**) 0% PEDOT: PSS, (**b**) 0.5% PEDOT: PSS, (**c**) 1.0% PEDOT: PSS, (**d**) 1.5% PEDOT: PSS and (**e**) 2.0% PEDOT: PSS hydrogels samples.

**Table 1 polymers-13-02901-t001:** List of samples and the amount of PEDOT: PSS required for each 20mL Cs–Gel–Agarose solution.

No.	Samples	Volume of PEDOT: PSS in 20 mL Cs–Gel–Agarose Solution
1.	Cs–Gel–Agarose	0.00 mL
2.	Cs–Gel–Agarose–0.5% PEDOT: PSS	0.15 mL
3.	Cs–Gel–Agarose–1.0% PEDOT: PSS	0.30 mL
4.	Cs–Gel–Agarose–1.5% PEDOT: PSS	0.45 mL
5.	Cs–Gel–Agarose–2.0% PEDOT: PSS	0.60 mL

**Table 2 polymers-13-02901-t002:** List of samples with their respective average pore diameters.

No.	Samples	Average Pore Diameter (µm)
1.	Cs–Gel (control)	1658.90
2.	Cs–Gel–0.5% PEDOT: PSS	703.40
3.	Cs–Gel–1.0% PEDOT: PSS	872.30
4.	Cs–Gel–1.5% PEDOT: PSS	916.21
5.	Cs–Gel–2.0% PEDOT: PSS	837.68

**Table 3 polymers-13-02901-t003:** The electrical conductivity of solution samples without the addition of agarose in S cm^−1^ and S m^−1^ units.

Samples	Conductivity (S cm^−1^)
Pure PEDOT: PSS	1.71 × 10^−5^ ± 1.00 × 10^−7^
DMSO–PEDOT: PSS	3.75 × 10^−1^ ± 5.55 × 10^−3^
Cs–Gel	1.21 × 10^−5^ ± 5.77 × 10^−8^
Cs–Gel–0.5% PEDOT: PSS	1.24 × 10^−5^ ± 0.00
Cs–Gel–1.0% PEDOT: PSS	1.33 × 10^−5^ ± 5.77 × 10^−8^
Cs–Gel–1.5% PEDOT: PSS	1.33 × 10^−4^ ± 1.53 × 10^−7^

**Table 4 polymers-13-02901-t004:** The electrical conductivity of solution samples with the addition of agarose in S cm^−1^ units.

Samples	Conductivity (S cm^−1^)
Cs–Gel	1.21 × 10^−5^ ± 5.77 × 10^−8^
Cs–Gel–Agarose	3.23 × 10^−4^ ± 5.51 × 10^−6^
Cs–Gel–Agar–0.5% PEDOT: PSS	3.27 × 10^−4^ ± 5.77 × 10^−8^
Cs–Gel–Agar–1.0% PEDOT: PSS	3.35 × 10^−4^ ± 1.53 × 10^−6^
Cs–Gel–Agar–1.5% PEDOT: PSS	3.28 × 10^−4^ ± 6.51 × 10^−6^
Cs–Gel–Agar–2.0% PEDOT: PSS	3.26 × 10^−4^ ± 1.00 × 10^−6^

## Data Availability

Not applicable.

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
