# Peer review of "Synthesis and Characterization of Porous, Electro-Conductive Chitosan–Gelatin–Agar-Based PEDOT: PSS Scaffolds for Potential Use in Tissue Engineering"

_polymers, 2021, doi:10.3390/polym13172901_

Round 1

Reviewer 1 Report

Manuscript entitled “Synthesis and Characterization of Electro-Conductive Chi- tosan-Gelatin based PEDOT: PSS Scaffolds for the Potential Use in Tissue Engineering” is a very interesting paper describing the use of Chitosan-gelatin for potential use in tissue engineering. However, the manuscript should be improved before being considered for publication in Polymer:

  1. In general, English language and style must be improve in the whole manuscript, mainly in sections such as the introduction and the experimental section
  2. Authors need to draw schematic or graphical abstract to summarize this manuscript, it would be helpful for readers.
  3. In the introduction, the background about the hydrogel, the authors should enrich this part and emphasize the necessity of the hydrogel by citing some recent literatures: https://www.mdpi.com/1999-4923/10/1/16, https://pubs.rsc.org/en/content/articlelanding/2019/bm/c9bm00139e/unauth, https://pubs.rsc.org/en/content/articlelanding/2020/bm/d0bm01004a/unauth
  4. What about stability of porous hydrogel?
  5. How authors can justify flow and rheological properties  of porus hydrogel?
  6. This porus hydrogel will shown immune response, how it can be evade?
  7. In vitro characterization study is too little for porus hydrogel, authors need to perform more characterization experiments.
  8. Is this system is biocompatible in vivo and in vitro, If yes then how?
  9. Discussion part should be improve and compare with already published literature.

Author Response

Dear Reviewer,

Many thanks for your feedback and comments. We truly appreciate them.

Thank you

Reviewer 2 Report

The work deals with the fabrication of conductive scaffold materials for potential use in hard tissue application. The authors prepared various scaffolds containing chitosan/gelatin combined with a various % percentage of PEDOT/PSS to optimize its conductivity for the cortical bone tissue. The paper is well arranged, experimental stages were given clearly and supported with related results. However, some minor corrections are needed before publication as follows:

C1) Section 3.3. : “As il-22 lustrated from the graph of Figure 14, the intensities of peaks from…” There is no Figure 14 in the manuscript. Please change it with the related one.

C2) Section 3.3 : “3.3. Electrical Conductivity of Porous Hydrogels” This is supposed to be section 3.4, since Section 3.3. belongs to FTIR results.

C3) Section 3.3 : “Electrical Conductivity of Porous Hydrogels “ . The first sentence of “Electrical conductivity of hydrogel solutions was measured using HIOKI 3520 LCR 38 Hi-Tester Electrochemical Impedance Spectroscopy (EIS).” Is already given in the experimental section. It would be better to remove it.

C4) Table 2 : Both conductivity units were given for S m-1 . Please correct one of those.

C5) Section 3.2 :  This section is mostly related to the introduction and experimental part, rather than results. It would be better to reconsider each paragraph that is not related to results and move it to the introduction and experimental section. Furthermore, The subsection “ (a)  Possible Reaction of Chitosan-Gelatin-Agar based PEDOT: PSS Porous Hydrogels Scaffolds” needs to be 3.2.1, since lack of the following subsections of b), c), etc.

Author Response

(The authors gave the same response as above.)

Reviewer 3 Report

Summary 

The manuscript entitled “Synthesis and Characterization of Electro-Conductive Chitosan-Gelatin based PEDOT: PSS Scaffolds for the Potential Use in Tissue Engineering’’ by Ruzaidi et al. shows the development characterization of electro-conductive chitosan-gelatin (Cs-Gel) based PEDOT: PSS scaffolds.

General comments

In general, the work is not as extremely accurate and well presented as a Polymers articles should be, but the presented results are certainly of interest to readers of this journal. The article style is not correct, but it should be reviewed in a few points. I believe that the text needs some technical adjustments to be published. Therefore, I recommend that this manuscript can be published in Polymers after major revision.

Specific comments

Going into detail on the specific issues, here some comments are reported:

- the article's grammar, punctuation, and style are not adequate, and the manuscript needs to be deeply proofread.

- there is a problem in the Introduction section, indeed a few vital notions should be given to readers. Conductive polymer-based scaffolds are mainly used for neural tissue engineering [https://doi.org/10.1021/acs.biomac.1c00524] and cardiac tissue engineering [https://doi.org/10.1016/j.biomaterials.2021.121008]. This should be highlighted in the Introduction citing the two reported brand new papers published in impactful journals.

- it is very odd to see a manuscript in which the main claim is the development of biomaterials, with no bio-related tests at all. Ruzaidi et al. should provide the results of the full biocompatibility test (confocal, imaging MTT, etc etc) otherwise the focus of the manuscript should be revised completely, removing any claim connected to the development of biomaterials from every section of the manuscript (e.g., Title, Abstract, Introduction, Conclusion).

- DMSO is cytotoxic at 3%. Even if it is used to dope a material that in turn will be applied as a filler, the use of it is very questionable and its effect on cells should be evaluated.

- Ruzaidi et al. should provide at least SEM, FT-IR and XRD for every fabricated material,

- SEM images. Every machine-generated piece of information (e.g. mag) should be removed. The visibility of all the scale bars should be improved.

- Error bars (standard deviation) should be added in each point reported in Figure 11.

Conclusion

The topic of this manuscript falls within the scope of Polymers. I like the concept proposed in this paper, anyway I think the manuscript needs a few improvements. I believe the article is of sufficient quality and novelty to meet the Polymers publication standard after a major revision.

Author Response

(The authors gave the same response as above.)

Round 2

Reviewer 1 Report

Accepted in current form.

Author Response

We appreciate the reviewer's feedback to accept the manuscript. Thank you.

Reviewer 3 Report

The article has been strongly improved and It can be accepted in the present form.

Author Response

We appreciate the reviewer's feedback to accept the manuscript in the present form.

Thank you.